# The Prevention of Periprosthetic Joint Infection in Primary Total Hip Arthroplasty Using Pre-Operative Chlorhexidine Bathing

**DOI:** 10.3390/jcm10030434

**Published:** 2021-01-23

**Authors:** Wen-Chi Su, Yu-Chin Lai, Cheng-Hung Lee, Cheng-Min Shih, Chao-Ping Chen, Li-Ling Hung, Shun-Ping Wang

**Affiliations:** 1Department of Nursing, Taichung Veterans General Hospital, Taichung 40705, Taiwan; wenchi0919@gmail.com (W.-C.S.); vghtc3760@gmail.com (Y.-C.L.); llhungch9177@gmail.com (L.-L.H.); 2Department of Orthopedics, Taichung Veterans General Hospital, Taichung 40705, Taiwan; 298f@vghtc.gov.tw (C.-H.L.); 10chengmin@gmail.com (C.-M.S.); cpchen@vghtc.gov.tw (C.-P.C.); 3Department of Food Science and Technology, HungKuang University, Taichung 43302, Taiwan; 4Department of Physical Therapy, HungKuang University, Taichung 43302, Taiwan; 5Department of Biological Science and Technology, National Chiao Tung University, Hsinchu 30010, Taiwan; 6Jen-Teh Junior College of Medicine, Nursing and Management, Miaoli County 35664, Taiwan; 7Department of Nursing, HungKuang University, Taichung 43302, Taiwan; 8Sports Recreation and Health Management Continuing Studies-Bachelor’s Degree Completion Program, Tunghai University, Taichung 40704, Taiwan

**Keywords:** surgical site infection, chlorhexidine, bathing, total hip arthroplasty, primary THA, periprosthetic joint infection, PJI

## Abstract

Periprosthetic joint infection (PJI) after total hip arthroplasty (THA) is a devastating complication. The aim of this study was to investigate whether preoperative bathing using chlorhexidine gluconate (CHG) before THA can effectively reduce the postoperative PJI rate. A total of 933 primary THA patients, with the majority being female (54.4%) were included in the study. Primary THA patients who performed preoperative chlorhexidine bathing were assigned to the CHG group (190 subjects), and those who did not have preoperative chlorhexidine bathing were in the control group (743 subjects). The effects of chlorhexidine bathing on the prevention of PJI incidence rates were investigated. Differences in age, sex, and the operated side between the two groups were not statistically significant. Postoperative PJI occurred in four subjects, indicating an infection rate of 0.43% (4/933). All four infected subjects belonged to the control group. Although the PJI cases were significantly more in the control group than in the CHG group, statistical analysis revealed no statistical significance in the risk of PJI occurrence between the two groups (*p* = 0.588). Preoperative skin preparation by bathing with a 2% chlorhexidine gluconate cleanser did not produce significant effects on the prevention of postoperative PJI in primary THA.

## 1. Introduction

Total hip arthroplasty (THA) is one of the most successful reconstructive procedures in orthopedic surgery today. It is indicated for end-stage hip osteoarthritis due to primary causes, inflammatory disease, osteonecrosis, post-traumatic hip fractures, and residual metastatic tumor or pyogenic hip arthritis. THA has relatively high survivorship and patient-reported satisfaction [1,2,3]. However, surgical site infection (SSI) or periprosthetic joint infection (PJI) after THA is a devastating complication that may cause the revision of THA, impairment in daily function and quality of life, and poor outcomes after THA [4,5]. Post-THA infection rates range from 0.4 to 1.4%, with most infections occurring within one year postoperatively [5,6,7,8]. Although the incidence rate of post-THA PJI is relatively low, infections are significantly associated with patient morbidity and mortality, and often require revision surgery, which leads to increased surgical risk [9]. Once PJI occurs after THA, the financial burden of revision procedures is quite high [10,11]. Therefore, the prevention of PJI after THA is of great importance.

Various infection prevention strategies have been developed for reducing the risk of SSI or PJI in clinical settings [12], including prophylactic antibiotics, antibiotic-loaded bone cement, laminar flow systems, optimal patient selection, and chlorhexidine application [8,13,14]. Chlorhexidine is a broad-spectrum biocide effective against Gram-positive and Gram-negative bacteria. It is widely used in clinical practice for preoperative skin cleansing [15], surgical site preparation [16], surgical hand antisepsis [17], and skin disinfection at central venous catheter insertion sites [18]. Therefore, the National Nosocomial Infections Surveillance System recommends routine skin cleansing with chlorhexidine the night before surgery to reduce the incidence of SSI [19]. In recent years, the uses of different chlorhexidine-based agents such as pre-operative cloth draping or bathing, peri-operative skin preparation, or intraoperative irrigation have become more popular and have been suggested for the prevention of infection after total joint replacement (TJR) [20,21,22,23,24,25].

Previous research has asserted that the use of chlorhexidine reduces bacterial colonization of the skin, thereby reducing the incidence rate of SSI. The effectiveness of chlorhexidine cloth use in reducing PJI incidence after TJR has also been demonstrated. However, the TJR cases analyzed in many studies included patients who had undergone either total knee and hip replacement or primary and revision TJR and has resulted in considerable heterogeneity. Therefore, it is difficult to ascertain the effects of chlorhexidine-based agents on the prevention of PJI in primary THA patients based on the results of previous studies. At present, the strength of the evidence for preoperative chlorhexidine bathing for the prevention of PJI after primary THA remains uncertain, and there is a lack of consistent suggestion and consensus in the relevant literature [26,27,28,29]. To the best of our knowledge, this comparative study is the first to specifically investigate the effects of preoperative chlorhexidine bathing on reducing the incidence rate of PJI after primary THA.

The purpose of this retrospective study was to assess the effects of bathing with a 2% chlorhexidine gluconate (CHG) cleanser the night before surgery to prevent PJI after primary THA.

## 2. Materials and Methods

### 2.1. Study Participants

A retrospective study design was adopted. Prior to commencement, this study was approved by the Institutional Review Board (IRB) of our institution (No.: CE20297A). Data used in the study were retrieved from the medical records of patients who underwent total hip replacement at our institution. To prevent excessive heterogeneity in the enrolled cases, we excluded patients aged <18 years as well as those with a history of tumor prosthesis replacement after tumor resection, THA followed by failure after fracture fixation, hemophilia, and prior infection of ipsilateral hip. Only patients who underwent primary THA were included as subjects for the comparison of the effects of preoperative chlorhexidine bathing on the postoperative incidence of PJI at the surgical site. Based on data from the medical records and the database of the infection control system, 1613 patients who underwent hip joint replacement between Jan 2015 and Dec 2018 were preliminarily enrolled in the study; of these, 482 patients who underwent hemi-arthroplasty and 185 patients who underwent revision THA were excluded. Among the remaining primary THA patients, six patients who had infections before THA, 2 patients with hemophilia, and five patients who had previously undergone open reduction and internal fixation after hip fracture were excluded. The last surveillance date of PJI in our infection control system was set on 31 December 2019, at least one year since the day of the leading THA. The mean follow-up time in this cohort is 36.6 ± 14.1 (12.3–60.7) months from the day of surgery to the last surveillance date. Finally, 933 primary THA patients were included in the study, as shown in Figure 1. The indications of these primary THA cases included osteoarthritis: 482 (51.66%), rheumatoid arthritis: 5 (0.54%), ischemic necrosis of the femoral head (INFH): 419 (44.91%), post-traumatic OA of the hip: 15 (1.61%) and developmental dysplasia of the hip (DDH): 12 (1.29%). Most of the implanted primary THA cases enrolled in this study were cementless. Of 933 THA cases, only 42 cases are cemented, and the other 891 cases are non-cemented.

### 2.2. Anti-Infective Protocol

The same preoperative skin preparation, perioperative disinfection, and postoperative anti-infection protocols were applied to all patients. Prophylactic systemic intravenous antibiotics were given 1 h before surgery, according to Centers for Disease Control and Prevention recommendations. All primary THA cases in this study underwent the same multimodal anti-infective protocol that was standardized at our institution and similar to those used in the previous study [30]. In our institution, the standard qualified operation theatres with laminar airflow are available. In operation theatres, the surgical site was initially washed with 2% chlorhexidine gluconate detergent as clean as possible, and then the skin was disinfected using an Iodine tincture solution at least two times before draping. Preoperative weight-based antibiotic prophylaxis covering gram(+) bacteria was given to the patient within 60 min prior to THA. The use of antibiotic cement for THA was at the preference of the surgeon.

Easy Antiseptic Cleansing Solution 2%^®^ (Panion & BF Biotech Inc., Taipei, Taiwan) approved by the Taiwan Food and Drug Administration FDA (No.: 057984) is the 2% chlorhexidine gluconate solution mixed with detergent used in cases of CHG groups, one night before surgery in this study. According to the manufacturer’s recommendations, the chlorhexidine cleanser should be applied to all parts of the body and remain in contact with the skin for at least 30 s before rinsing and repeating this procedure two times or more. Before providing the chlorhexidine cleanser to patients, healthcare personnel at our institution provided adequate patient education and clear instructions to guide patients on the proper use of the cleanser.

### 2.3. The Definition of PJI

Table 1 shows the diagnostic criteria for PJI after THA [31]. The 933 primary THA patients were divided into two groups based on the preoperative bathing procedure: the control group consisted of 743 subjects who adopted routine bathing using soap and water the night before surgery, and the CHG group consisted of 190 subjects who bathed using 2% chlorhexidine gluconate solution in addition to routine cleansing with soap and water the night before surgery.

### 2.4. Statistical Analysis

Statistical analysis was performed on data retrieved from the subjects’ medical records and the database of the infection control system. Continuous data, such as patient age, were expressed as the mean and standard deviation; categorical data, such as patient sex, the operated side, and the occurrence/non-occurrence of postoperative infection, were expressed in numbers and percentages. Differences in continuous data between the control group and CHG were assessed using the Mann–Whitney U test, and the categorical data of the two groups were compared using Fisher’s exact test and the Chi-squared test. All statistical analyses were performed using the Statistical Package for Social Science (IBM SPSS version 22.0; International Business Machines Corp, New York, NY, USA). The level of the statistical significance was set at *p* < 0.05.

## 3. Results

A total of 933 subjects were included in this study, with 508 (54.4%) being female. The operated side was the right hip in 510 subjects (54.7%), and the left hip in 423 subjects (45.3%). The control group included 743 subjects (396 females (53.3%)) with a mean age of 59.63 ± 15.42 years; the operated side was the right hip in 398 subjects (53.5%) and the left hip in 345 subjects (46.4%). CHG group included 190 subjects (112 females (58.9%)) with a mean age of 57.71 ± 15.23 years; the operated side was the right hip in 112 subjects (58.9%) and the left hip in 78 subjects (41.0%). In both groups, there were more females than males, and the right hip was operated on more often than the left. Differences in age, sex, and the operated side between the two groups were not statistically significant (Table 2).

Postoperative infection occurred in four of the 933 primary THA patients, indicating a total infection rate of 0.43%. All cases of PJI occurred within 90 days postoperatively, with the duration of infection ranging from 21 to 54 days (mean: 36.7 days). The pathogens of the four infections and each onset time after THA are listed in Table 3.

All the four infected subjects belonged to the control group, that is, they had not used the chlorhexidine cleanser the night before surgery, and PJI incidence within the control group was 0.54% (4/743). There were no cases of PJI in the CHG group. Although the number of PJI cases was higher in the control group than the CHG group (control vs. CHG = 4:0), an analysis of the risk of PJI using Fisher’s exact test revealed that the difference between the two groups was statistically insignificant (*p* = 0.588). This indicates that there was no significant difference in infection rate between patients who used the chlorhexidine cleanser for preoperative skin preparation and those who did not (Table 4). Therefore, our results demonstrate that the use of a 2% chlorhexidine cleanser had no significant effects on post-THA SSI.

## 4. Discussion

Among the subjects included in this study, there were more females than males, and the mean ages of subjects in the control and CHG groups were 59.6 years and 57.7 years, respectively. Of the 743 control subjects, four developed PJI, indicating a PJI incidence rate of 0.54% in the control group. Although this incidence rate was significantly higher in the control group than that of the CHG group (0%), there was no significant statistical difference in the risk of PJI between the two groups. Such a result is in agreement with certain studies that have indicated the lack of conclusive evidence that chlorhexidine bathing reduces the risk of SSI [27,29] but is inconsistent with the results reported by Kapadia et al., who demonstrated that cleansing with 2% chlorhexidine cloth the night before and on the morning of surgery, reduced the risk of SSI in TJR patients [20,25,32].

Despite a large amount of literature supporting the view that the use of chlorhexidine reduces the incidence of PJI after TJR, evidence that preoperative chlorhexidine-based skin cleansing prevents infection after TJR is less convincing. The subjects of these studies included patients who underwent total knee arthroplasty (TKA) and THA, or patients who received primary TKA/THA and revision surgery, which makes it difficult to ascertain the true effects of chlorhexidine on the incidence of PJI after primary THA. In addition, preoperative cleansing in these studies was performed by wiping with chlorhexidine cloth, which differs from the chlorhexidine bathing method adopted in the present study [25,26,27,33]. We believe that the different methods of usage of pre-operative chlorhexidine-based agents can also lead to different effects on the prevention of PJI after TJR, but further studies are required to compare the preventive effects of preoperative chlorhexidine cleansing by different methods on PJI after TJR. As this study was focused solely on investigating the effects of preoperative chlorhexidine bathing, our results can serve as a key reference for assessing the adoption of chlorhexidine bathing in clinical care for the prevention of PJI after primary THA.

The prevention of PJI after THA is very crucial to avoid devastating sequelae that result in a higher rate of revision surgeries. A multi-modal standardized disinfected protocol is used in our institution. The varied clinical applications of chlorhexidine gluconate in total joint arthroplasty including preoperative impregnated cloth, bathing cleanser, and hand antisepsis solution are common and easy to be applied in clinical practices and also convenient to the patient. Preoperative CHG showers can effectively reduce SSIs in surgically treated cases and central line-associated bloodstream infections in comparison with those who did not use them in previous reports [34,35]. However, the efficacy of chlorhexidine-based agents in the prevention of PJI for total hip arthroplasty is still inconclusive. Kapadia et al. suggested that the preoperative chlorhexidine wiping significantly decreased the PJI rate on the total hip arthroplasty. However, mixed enrollment with THA and revision THA, [24,25] or primary and revision TKA with THA [20] was noticed in all of these studies. This will result in heterogeneity of the studies hence there are different infection rates in these varied total joint arthroplasties. Furthermore, our finding in this current study suggesting no significant effect of the PJI prevention of primary THA is incompatible with the previous reports. This present research is the first study elucidating the efficacy of chlorhexidine bathing on the PJI prevention of primary THA.

According to the manufacturer’s recommendations, the chlorhexidine cleanser should be applied to all parts of the body and remain in contact with the skin for at least 30 s before rinsing. Before providing the chlorhexidine cleanser to patients, healthcare personnel at our institution provided adequate patient education and clear instructions to guide patients on the proper use of the cleanser. However, improper chlorhexidine bathing techniques can cause insufficient contact duration and concentration of chlorhexidine during bathing, which may affect its disinfectant effects [36]. Although Kapadia et al. reported that wiping with chlorhexidine cloth before TJR reduced the risk of SSI in patients [25], our results showed that chlorhexidine bathing the night before surgery had no significant effects on reducing infection risk after primary THA. This may be partly due to poorer patient compliance with chlorhexidine bathing than with the use of chlorhexidine cloth, which may have limited its effects on preventing post-TJR infection.

This study has a number of limitations. First, the retrospective nature of the study may have led to certain unknown biases and variability in data collection. Next, the study had a limited sample size as it was focused solely on primary THA patients, which may have caused the results of this study to be underpowered. Therefore, future prospective studies with a larger sample size may be needed to validate the findings of this study. Third, the absence of analysis of wound complications and patient comorbidities also should be considerable limitations. Lastly, the uncertain compliance of chlorhexidine cleanser used in pre-operative bathing was also a limitation of this study. Although we provided adequate patient education and clear instructions to each patient on the use of the chlorhexidine cleanser, we were unable to ensure that the cleanser was properly used by all the patients. Furthermore, the absence of supervision if the patient had performed the bathing or not is also a limitation. Consequently, the disinfectant effects of the cleanser may have been limited by improper use.

## 5. Conclusions

In this study, we investigated the effects of preoperative chlorhexidine bathing on the prevention of PJI in THA patients. Patients who underwent hip joint replacement were preliminarily enrolled, and cases that could result in significant heterogeneity were excluded to maintain consistency among the study subjects. We found that preoperative skin preparation by bathing with 2% chlorhexidine gluconate cleanser had no significant effects on the prevention of PJI after primary THA. The results of this study can serve as a reference for surgeons to assess if chlorhexidine bathing should be regularly performed before primary THA. However, further large-scale prospective studies are required for the validation of our results.

## Figures and Tables

**Figure 1 jcm-10-00434-f001:**
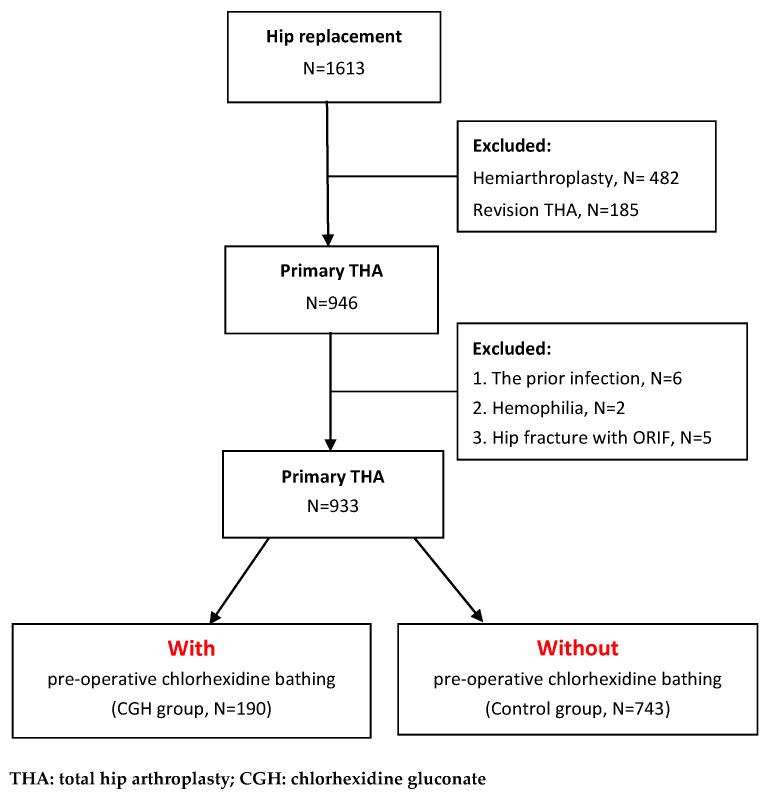
The flowchart of primary THA case enrollment.

**Table 1 jcm-10-00434-t001:** Definition of Periprosthetic Join Infection * [31]. Reproduced with permission from Javad Parvizi and Thorsten Gehrke, The Journal of Arthroplasty; published by Elsevier, 2014.

Major Criteria	Two positive periprosthetic cultures with phenotypically identical organisms, OR
	A sinus tract communicating with the joint, OR
Minor Criteria	(1) Elevated serum C-reactive protein (CRP) AND erythrocyte sedimentation rate (ESR)
	(2) Elevated synovial fluid white blood cell (WBC) count OR ++change on leukocyte esterase test strip
	(3) Elevated synovial fluid polymorphonuclear neutrophil percentage (PMN%)
	(4) Positive histological analysis of periprosthetic tissue
	(5) A single positive culture

* PJI is present when one of the major criteria exists or three out of five minor criteria exist.

**Table 2 jcm-10-00434-t002:** The characteristics of primary THA cases.

	CHG (N = 190)	Control (N = 743)	*p*-Value
Age (mean ± SD)	57.71 ± 15.23	59.63 ± 15.42	0.170
Sex (N (%))			0.189
Female	112 (58.9%)	396 (53.3%)	
Male	78 (41.0%)	347 (46.7%)	
Laterality (N (%))			0.212
Rt side	112 (58.9%)	398 (53.5%)	
Lt side	78 (41.0%)	345 (46.4%)	

Mann–Whitney U test. Chi-Square test. * *p* < 0.05, ** *p* < 0.01.

**Table 3 jcm-10-00434-t003:** The pathogens and onset time of the four infections.

Case No.	Pathogens	Onset time (Days)	Cemented
1	Staphylococcus aureus	54	No
2	Acinetobacter baumannii	43	Yes
3	Pseudomonas aeruginosa	21	No
4	Staphylococcus epidermidis	30	No

**Table 4 jcm-10-00434-t004:** The periprosthetic infection rate of primary THA between groups.

	CHG (N = 190)	Control (N = 743)	Total (N = 933)	*p*-Value
PJI cases	0 (0%)	4 (0.54%)	4 (0.43%)	0.588
Within 90 days	0 (0%)	4 (0.54%)	4 (0.43%)	
Beyond 90 days	0 (0%)	0 (0%)	0 (0%)	
No PJI cases	190 (100%)	739 (99.46%)	929 (99.57%)	

Fisher’s Exact Test. * *p* < 0.05, ** *p* < 0.01.

## Data Availability

The data presented in this study are available from the corresponding author.

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
