# Peer review of "The Prevention of Periprosthetic Joint Infection in Primary Total Hip Arthroplasty Using Pre-Operative Chlorhexidine Bathing"

_jcm, 2021, doi:10.3390/jcm10030434_

Round 1

Reviewer 1 Report

The authors present their results of the use of chlorhexidine bath before Primary THA compared to patients who underwent THA; however, without the use of chlorhexidine bath. This Topic has been indeed addressed in previous numerous studies. In spite of the retrospective design, the inclusion of only Primary cases and only THA procedures in the current work utlizing the bathing method might be of value.

A total of 0,4% PJI rate is relatively low, what was the follow-up time?

What were the pathogens of the 4 infections and the time of each?

What about wound complications?

Were the implanted THA cemented or cementless?

You mentioned, as preventive measures, the skin preparation in the investigated group and the antibiotic Prophylaxis in both groups, do you applied further preventiv measures to reduce the Risk of PJI? Schould be clarified

What was the indications in detailes for Performing of THA in both Groups?

Patient's comorbidities (Risk scores) should be included

Introduction has to be shortend

Detailed description about chlorhexidine should be added to the method section

Who manufactured the investigated chlorhexidine?

Did the control group have any skin preparations before THA?

The absence of supervision or Control if the Patient had performed the bathing or not is a limitation, what is your comment?

These results have to be compared with this paper: Does Preadmission Cutaneous Chlorhexidine Preparation Reduce Surgical Site Infections After Total Hip Arthroplasty? Kapadia BH, Jauregui JJ, Murray DP, Mont MA.Kapadia BH, et al. Clin Orthop Relat Res. 2016 Jul;474(7):1583-8. doi: 10.1007/s11999-016-4748-9.

Author Response

We are very thankful for your review and suggestion!

Reviewer 2 Report

Line 19-20 - and those who in control group (743 subjects) did not have preoperative chlorhexidine bathing  - Please rephrase as - "and those who did not have preoperative chlorhexidine bathing in to the control group (743 subjects)." 

Introduction is too long, needs to be concise.  Last paragraph talks about exclusion criteria, which can be moved to Material and methods.

Lines 77-78 - Needs to be in conclusion

Lines 158-160 - Although there were no cases of PJI in the CHG group, which led to a significantly higher number of PJI cases in the control group than the CHG group (control:CHG = 4:0).    - Please rephrase this

I think the discussion part is not adequate. I think the authors should discuss few more studies comparing their results with them.

Authors are presuming that one of the reasons could be noncompliance with the use of CHG. Though this is a possibility, I don't think this is relevant in this study because of none of the CHG group patients developed infection !

The main limitation of this study is the small sample size. No prior power analysis was done. With the incidence of PJI being very low, large sample sizes are needed to make strong recommendations and any definitive conclusions.

Author Response

We are very grateful for your review and suggestion !

Round 2

Reviewer 1 Report

Follow-up time must be exactly mentioned (mean, range, and SD).

The given indications for THA should be shortly mentioned.

The absence of Analysis of wound complication should be added as a limitation.

The absence of Analysis of patient's comorbidities should be added as a limitation.

Author Response

We are very grateful for the suggestion from the reviewers.

  1. Follow-up time must be exactly mentioned (mean, range, and SD).

Reply:

We are very grateful for the suggestion from the reviewers.

The last surveillance date of PJI in our infection control system was set on Dec. 31, 2019, at least one year since the day of the leading THA. The mean follow-up time in this cohort is 36.6 ± 14.1 (12.3 – 60.7) months from the day of surgery to the last surveillance date.

  1. The given indications for THA should be shortly mentioned.

Reply:

We are very grateful for the suggestion from the reviewers.

The given indications for THA has been added to Method section of the present study as below. “The indications of these primary THA cases included osteoarthritis: 482(51.66%), rheumatoid arthritis: 5(0.54%), ischemic necrosis of the femoral head (INFH): 419(44.91%), post-traumatic OA of hip: 15(1.61%) and developmental dysplasia of the hip (DDH): 12(1.29%).

  1. The absence of Analysis of wound complication should be added as a limitation.

Reply:

We are very grateful for the suggestion from the reviewers.

The change has been added to the third limitation of the present study.

  1. The absence of Analysis of patient's comorbidities should be added as a limitation.

Reply:

We are very thankful for the suggestion from the reviewers.

The change has been added to the third limitation of the present study.

Reviewer 2 Report

I appreciate the authors making all the suggested corrections to the manuscript.

Please correct the spelling as below 

Line 245 - However, the efficacy of chlorhexidine
based agents in prevention of PJI for total hip arthroplasty is still inclusive. - "inconclusive"

Author Response

Thanks for the suggestion from the reviewer !

Please correct the spelling as below

Line 245 - However, the efficacy of chlorhexidine

based agents in prevention of PJI for total hip arthroplasty is still inclusive. - "inconclusive"

Reply:

We are very grateful for the suggestion from the reviewers.

The correction of spelling has been made in the present manuscript.